# Drone-YOLO: An Efficient Neural Network Method for Target Detection in Drone Images

Zhengxin Zhang

College of Information Science and Technology, Zhongkai University of Agriculture and Engineering, Guangzhou 510225, China; zx_zhang@buaa.edu.cn

**Abstract:** Object detection in unmanned aerial vehicle (UAV) imagery is a meaningful foundation in various research domains. However, UAV imagery poses unique challenges, including large image sizes, small sizes detection objects, dense distribution, overlapping instances, and insufficient lighting impacting the effectiveness of object detection. In this article, we propose Drone-YOLO, a series of multi-scale UAV image object detection algorithms based on the YOLOv8 model, designed to overcome the specific challenges associated with UAV image object detection. To address the issues of large scene sizes and small detection objects, we introduce improvements to the neck component of the YOLOv8 model. Specifically, we employ a three-layer PAFPN structure and incorporate a detection head tailored for small-sized objects using large-scale feature maps, significantly enhancing the algorithm's capability to detect small-sized targets. Furthermore, we integrate the sandwich-fusion module into each layer of the neck's up–down branch. This fusion mechanism combines network features with low-level features, providing rich spatial information about the objects at different layer detection heads. We achieve this fusion using depthwise separable evolution, which balances parameter costs and a large receptive field. In the network backbone, we employ RepVGG modules as downsampling layers, enhancing the network's ability to learn multi-scale features and outperforming traditional convolutional layers. The proposed Drone-YOLO methods have been evaluated in ablation experiments and compared with other state-of-the-art approaches on the VisDrone2019 dataset. The results demonstrate that our Drone-YOLO (large) outperforms other baseline methods in the accuracy of object detection. Compared to YOLOv8, our method achieves a significant improvement in $mAP_{0.5}$ metrics, with a 13.4% increase on the VisDrone2019-test and a 17.40% increase on the VisDrone2019-val. Additionally, the parameter-efficient Drone-YOLO (tiny) with only 5.25 M parameters performs equivalently or better than the baseline method with 9.66M parameters on the dataset. These experiments validate the effectiveness of the Drone-YOLO methods in the task of object detection in drone imagery.

**Keywords:** object detection; drone image; remote sensing

## 1. Introduction

In the past 15 years, with the gradual maturation of drone control technology, UAV remote sensing imagery has become an important data source in the field of low-altitude remote sensing research due to its cost-effectiveness and ease of acquisition. During this period, deep neural network methods have been extensively researched and have gradually become the optimal approaches for tasks such as image classification [1–3], object detection [4–6] and image segmentation [7–9]. However, the majority of currently applied deep neural network models, such as VGG [1], RESNET [2], U-NET [7], PSPNET [8], have primarily been developed and validated using manually collected image datasets, such as VOC2007 [10],VOC2012 [11], MS-COCO [12], as shown in Figure 1.

These images are collected based on the photographer's subjective preferences. As shown in Figure 1: (1) many images are captured in small spaces scenes; (2) the number of objects in each image is relatively small, and the photographer will filter or avoid the

objects or backgrounds, which are irrelevant with the main object; (3) there are not too many densely arranged and overlapping targets in the image; (4) the image size is relatively small; (5) the ratio of the size of the target in the image to the size of the entire image is relatively large; (6) the image has good lighting conditions and no significant overexposure or insufficient light; (7) many of the image-shooting angles are heads-up, and there are few top-view images.

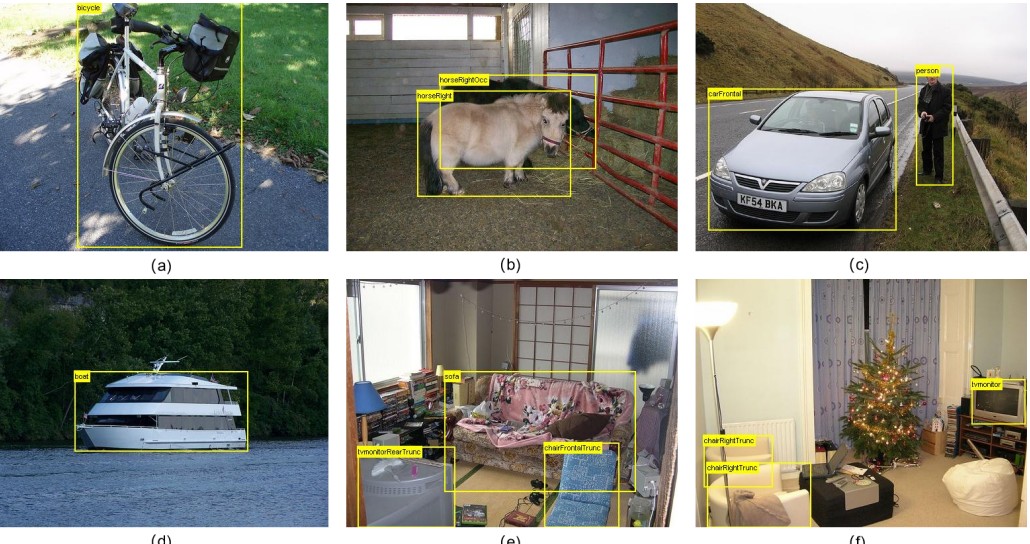

**Figure 1.** Sample images taken manually by people on the ground: (**a**) of a bicycle on a small road; (**b**) Ponies in the stable; (**c**) A car and a man on the roadside; (**d**) A boat in the river; (**e**) Objects in the bedroom; (**f**) Objects in the living room.

As shown in Figure 2, the images obtained from drones exhibit significant differences compared to manually captured ground-based imagery. These images taken by drones have the following characteristics: (1) most of the images are outdoor scenes; (2) the shooting angles of the images are mostly overhead views, with few upward perspectives; (3) the image scene is relatively large (outdoor images taken from at least 5 m above the ground); (4) there are many differently sized objects in the image. In many scenes, objects are densely arranged. In some scenes, many similar-looking objects are densely arranged and overlapped; (5) the image's background is complex; (6) the sizes of many images are large; (7) the ratio of the size of a single object to the size of the entire image is relatively small; (8) some images have illumination problems, such as glare, overexposure, and insufficient light.

In addition to these image data characteristics, there are two common application scenarios for drone remote sensing object detection methods. The first one involves post-flight data processing using large desktop computers. After the drone's flight, the captured data are processed on a desktop computer. The second one involves real-time processing during the flight, where an embedded computer on the drone synchronously processes the aerial imagery data in real time. This application is often used for obstacle avoidance and automated mission planning during drone flights. Therefore, the applied neural network object detection methods need to meet different requirements for each scenario. For methods applicable to desktop computer environments, high detection accuracy is required. For methods applicable to embedded environments, the model parameters need to be within a certain scale to meet the embedded hardware operational requirements. After meeting the operational conditions, the detection accuracy of the method also needs to be as high as possible.

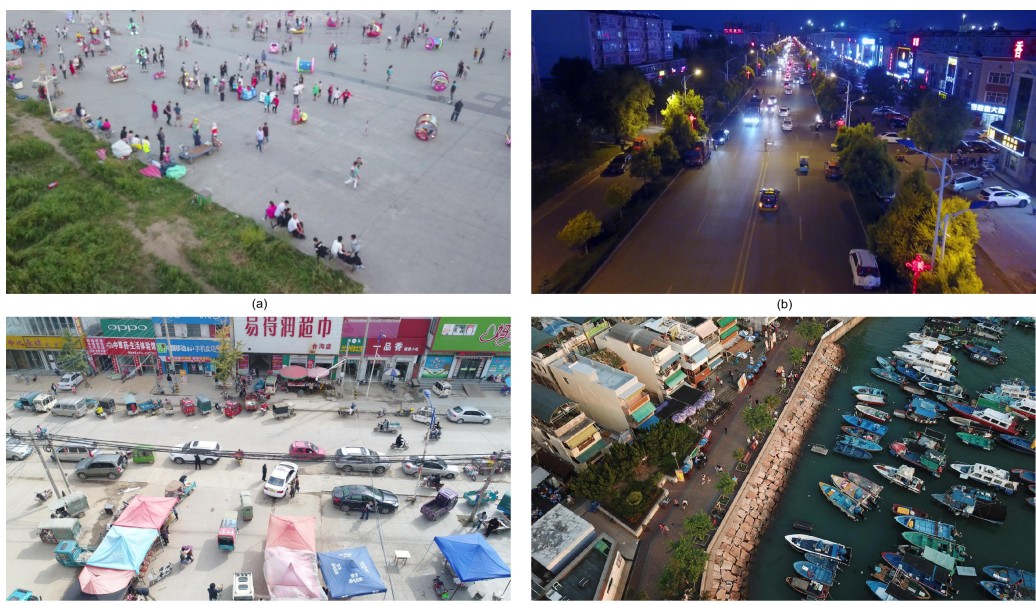

**Figure 2.** Sample images taken from drones: (**a**) Crowds on the square; (**b**) The traffic flow on the streets at night; (**c**) Setting up stalls and vehicles on daytime roads; (**d**) Buildings and docked ships along the coast;.

Therefore, neural network methods for object detection in drone remote sensing imagery need to be able to adapt to the specific characteristics of these data. They should be designed to meet the requirements of post-flight data processing, which can provide high accuracy and recall rates result, or they should be designed as models with smaller scale parameters that can be deployed in embedded hardware environments for real-time processing on drones.

In this work, we propose a series of improved neural network models based on the YOLOv8 architecture with unified improvements. Among them, the models with larger parameters can fulfill the requirements of post-flight data processing on desktop computers in terms of object detection accuracy and recall rate. On the other hand, the models with smaller scale parameters can meet the operational requirements of embedded hardware environments and achieve satisfactory detection accuracy.

The main contributions of the article include:

1. A series of novel neural network methods for multi-scale object detection in UAV images. These methods include high-accuracy models with larger parameters suitable for desktop platforms and smaller parameter models suitable for deployment on embedded edge computing devices. This provides a diverse range of effective options for different computing environments.
2. A sandwich-fusion module with a large receptive field and low model parameters, specifically designed for deployment in the neck network of the PAFPN structure. Spatial features from the lower layers of the backbone network were effectively captured by this module, leading to improved object detection performance.
3. The utilization of the RepVGG reparameterized convolution module to enhance the downsampling layers in the network backbone. This modification improved the effectiveness of the downsampling process, resulting in enhanced target detection performance.

Through ablation experiments, we demonstrate the feasibility and effectiveness of our proposed network optimization methods. Experimental results on the VisDrone2019 dataset show that our methods can improve the performances of the YOLOv8 series methods (large, medium, small, tiny, and nano) under limited net model parameter growth. Additionally, comparative analyses with state-of-the-art detection models and current mainstream models with adjacent parameters demonstrate the superiority of our proposed method.

The remaining sections of this paper are organized as follows: Section 2 provides a review of the related work, and Section 3 elaborates on our proposed method. Section 4 presents the experimental results, and the conclusions are presented in Section 5.

## 2. Related Work

In recent years, numerous object detection neural network methods, particularly one-stage object detection methods, have been proposed. Unlike the two-stage methods, the one-stage method combines object location detection and classification in one step, achieving real-time object detection on both desktop and embedded hardware. Among these works, the YOLO series [13–20] received much attention. In particular, since YOLOv4 [15] was proposed, many related works have been proposed. These methods not only achieve good detection results, but also offer a series of improvements in areas such as network model structures, activation functions, loss functions, network training methods, and training data augmentation methods.

In the research on densely distributed object detection, Chu et al. [21] proposed a neural network method, CrowdDet, for detecting dense and mutually occluded targets in images. The authors believe that the primary reasons for poor performance in dense object detection are (1) the presence of highly coincident instances (and their associated candidate boxes) in the image space that are likely to have very similar features, making it challenging for the detector to generate differentiated prediction results for each candidate box; (2) the severe overlap between instances, leading to prediction results being suppressed by non-maximum suppression (NMS) errors. To solve the above problem, the researchers proposed predicting a group of possibly highly coincident instance sets for each candidate box of the detection target, instead of predicting a single instance as usual. They introduced several methods: (1) EMD loss function, which is used to supervise the learning process of instance set prediction; (2) the set NMS post-processing method to suppress duplicate detection results of different candidate boxes and overcome the application defects of NMS method in dense scenes; (3) the optimization module for handling potential false positive targets. Bochkovskiy et al. [15] proposed YOLOv4. The author introduced the CSPDarkNet [22] as the backbone structure of the network, enhancing the learning abilities of convolution neural networks, allowing the network to maintain the accuracy of feature map extraction while being lightweight; reducing computing bottlenecks; and lowering memory costs. In the neck section of the network, the SPPF module was introduced and the PAFPN module [23] was adopted. Jocher et al. [16] proposed a one-stage target detection neural network model, YOLOv5. The method has high accuracy, low computational complexity, and high speed. In the beginning structure of the backbone, the author introduced the focus module. Compared with the convolutional layer with $3 \times 3$ kernel, this module has a higher computing speed, which mainly plays a role in speeding up the processing of the input image. The neck section adopts a PAFPN module. In the detection head section, an anchor-based method is deployed, and NMS is used to eliminate erroneous candidate boxes. Ge et al. [17] proposed YOLOX, a one-stage, object detection neural network model. The author still uses CSPDarkNet [22] as the backbone network, but introduces SiLU [24] as the activation function, which solves the problem of gradient dispersion when the input of the ReLU [25] function is negative and the output is 0. The author introduced decoupled heads, which separately implement confidence and regression boxes, and integrate them into one during prediction; the author introduced SimOTA to implement the anchor-free detector head. Li et al [18] proposed YOLOv6; EfficientRep [26] was introduced as the backbone of the network. Li et al. [27] proposed YOLOv6 3.0, which introduced the bidirectional concatenation (BiC) as the feature-map fusion module deployed in the PAFPN structure of the network's neck. Wang et al. [28] proposed ELAN, a network module designed by controlling the shortest and longest gradient paths. Wang at al. [19] proposed YOLOv7. The authors introduced an E-ELAN structure in the backbone of this work, which can train multi-scale features without increasing the

length of the shortest gradient path. Jocher et al. [20] proposed YOLOv8. The authors introduced the C2f module as the backbone stage module.

Mao et al. [29] proposed MSA-YOLO, an object detection method. The authors made improvements to the YOLOv5 model by introducing a multi-scale feature extraction module (MFEM), which combined the squeeze-and-excitation module (SEM) and spatial attention module (SAM) to form a hybrid attention mechanism known as the multi-scale split attention unit, serving as the backbone structure. In experiments compared with other network models on the VisDrone2019-test dataset, the authors achieved the best performance in terms of the $mAP_{0.5}$ metric. Among all 10 types of detected objects, MSA-YOLO obtained the best results in three types. Li et al. [30] proposed ACAM-YOLO, a modified version of YOLOv5 [16], with adaptive co-attention module (ACAM) mechanisms. Through experiments, the authors compared the difference in object detection results between the YOLOv5 network and the improved YOLOv5 using the ACAM. The experiment showed that the network's performance significantly improved, and the model parameters significantly reduced. Zhao et al. [31] proposed MS-YOLOv7. This network is improved based on YOLOv7 [19], increasing the original three detection heads to four, using the Swin transformer [32], W-MSA, SW-MSA, and the CBAM [33] attention mechanism to enhance the features of the network neck. Using the soft-NMS method improves the performance of the NMS method in densely distributed object detection. Li et al. [34] proposed an improved small parameter object detection network based on the YOLOv8-s model. The authors replaced the PAFPN structure of YOLOv8 with Bi-FPN [35] and improved the backbone using the Ghostblock [36] module, achieving a neural network method with fewer parameters but better detection performance. Liu et al. [37] proposed EdgeYOLO, which introduces a lite-decoupled head for object detection. In comparison with the decoupled head of YOLOX [17], the detection head with fewer parameters achieves a faster inference speed. Furthermore, it achieves comparable or superior accuracy compared to YOLOX.

Currently, one of the most significant challenges when applying these methods to UAV object detection is in effectively detecting small objects. UAV remote sensing images often have large image sizes, complex backgrounds, and a substantial presence of small objects. Our proposed solution focuses on optimizing the accurate detection of small objects. The advantage of YOLO series networks lies in their utilization of multi-level detection heads, enabling the detection of objects of various sizes from different levels of feature vectors. Our approach primarily revolves around the detection of tiny and small-sized objects using feature vectors from lower layers, which have a higher spatial resolution. To achieve this goal, we employ the sandwich-fusion module to optimize their semantic features. The detection heads can obtain feature vectors with both high spatial resolution and accurate semantic information, thereby enhancing the overall detection performance.

## 3. Methodology

Figure 3 shows the architecture of our proposed Drone-YOLO (large) network model. This network structure is an improvement on the YOLOv8-l model. In the backbone section of the network, we utilize the RepVGG structural re-parameterized convolution module as the downsampling layer. During training, this convolutional structure simultaneously trains both $3 \times 3$ and $1 \times 1$ convolutions. The two convolution kernels are merged into one $3 \times 3$ convolution layer during inference. This mechanism enables the network to learn more robust features without compromising inference speed or inflating the model size. In the neck section, we extend the PAFPN structure to three layers and append a small-size object detection head. By incorporating the proposed sandwich-fusion module, spatial and channel features are extracted from three different layer feature maps of the network's backbone. This optimization enhances the multi-scale detection head's ability to gather spatial positioning information of the objects to be detected.

We further refine the parameter size and model structure of the Drone-YOLO (large) network model to obtain the Drone-YOLO series of networks. Among them, the Drone-YOLO (tiny) and Drone-YOLO (nano) models have fewer parameters, yet achieve compa-

rable detection accuracy to recently proposed models. This network model series achieved good detection results on the VisDrone2019 dataset without utilizing widely employed attention mechanisms.

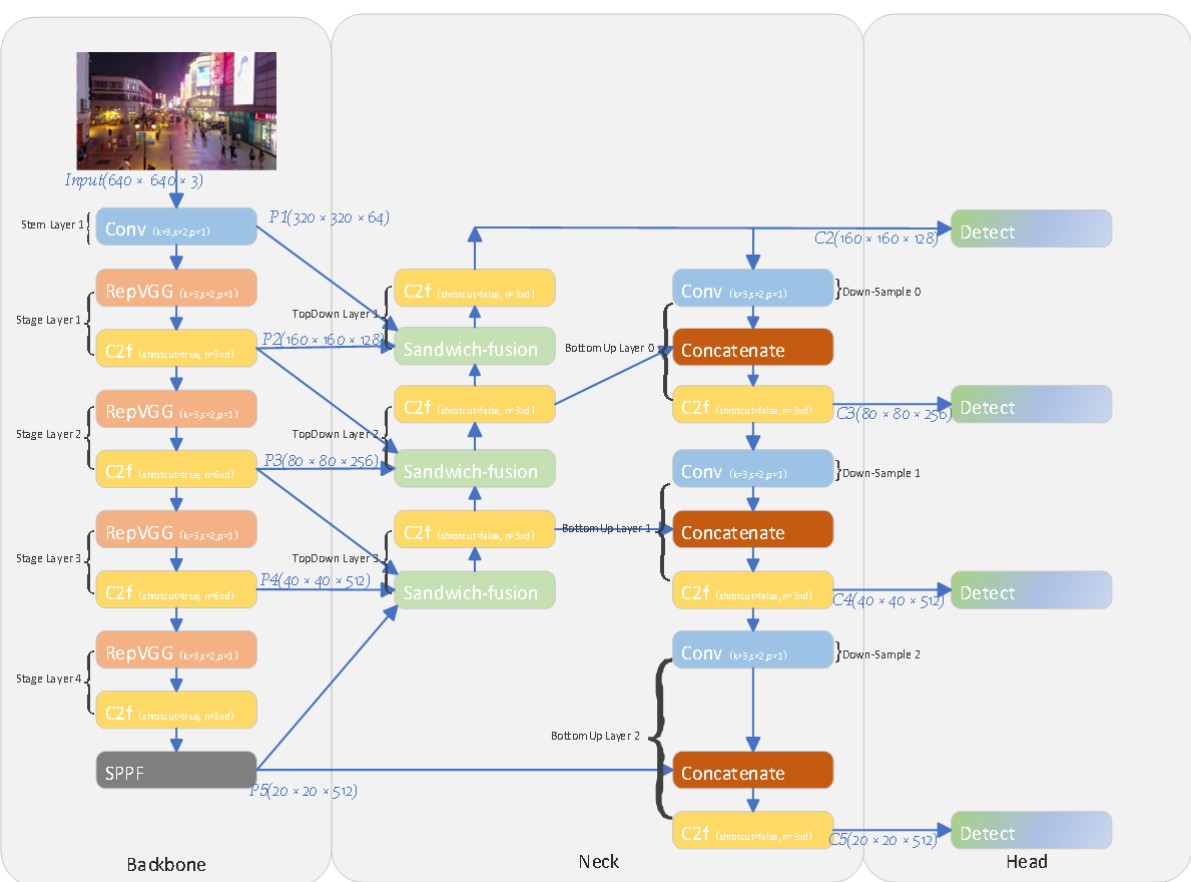

**Figure 3.** Proposed network model (large) with its backbone, neck and head.

### 3.1. Backbone

The backbone of our proposed network model includes a stem layer and four stage blocks. As shown in Figure 3, the stem layer is a convolutional layer with a stride of 2. Each stage block has a RepVGG block with a stride of 2, and is used as a downsampling module and a C2f convolutional block component, as proposed in YOLOv8.

In the proposed network, the feature map extraction backbone's essential multi-stage components are the C2f modules, which were modified from the C3 modules. In the C3 module of CSPDarkNet, the backbone of YOLOv5, the cross-stage partial structure was implemented and obtained good results. Its design goal is to reduce the network's computational complexity and enhance gradient performance. The design concept of DenseNet [3] inspired the C3 module's structure. The C3 module is composed of three convolutional layers with a $1 \times 1$ kernel and a DarkNet bottleneck, which has a sequence of convolutional layers with a $3 \times 3$ kernel. The module's input feature map is divided into two parts: One does not pass through the bottleneck, and the other passes through the bottleneck. In the part of the feature map used to compute through the bottleneck, the first layer's output result is concatenated with its input as the input of the second convolutional layer. The feature map passed through the bottleneck concatenates the other part feature map, which only computes through one $1 \times 1$ convolution layer and passes through a final $1 \times 1$ convolutional layer to be the output feature of the C3 module.

In the backbone of this proposed network, we use the C2f structure from the YOLOv8 network as each stage unit of the backbone. As shown in Figure 4, compared with the C3 structure, the C2f module sufficiently meets the idea proposed by the ELAN module [28],

which optimizes the network structure from the perspective of controlling the shortest and longest gradient paths, thereby making the network more trainable. In addition, without changing the network's shortest and longest gradient paths, the C2f module can effectively learn multi-scale features and expand the range of receptive fields through feature vector diversion and multi-level-nested convolution within its module.

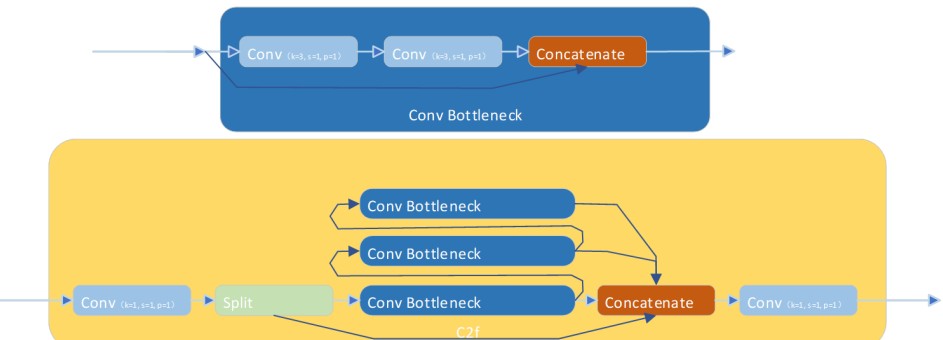

**Figure 4.** C2f convolutional block structure of YOLOv8.

We deployed RepVGG [38] blocks as the downsampling structure between stages in the backbone network. The RepVGG component is improved from the VGG [1], which is an early proposed single path network characterized by the extensive use of $3 \times 3$ convolutions as its essential units. The VGG is a memory-saving model because it does not require memory for identity structures like residual blocks in ResNet [2]. The $3 \times 3$ convolutional layer is a high-efficiency network structure. Machine learning hardware acceleration libraries, such as cudnn [39] and MKL, have significantly optimized the computational efficiency of $3 \times 3$ convolution, making the calculation density (theoretical FLOPs/time usage) [38] of $3 \times 3$ convolution reach four times that of $1 \times 1$ or $5 \times 5$ convolution. Consequently, extensive utilization of this type of convolutional neural network offers efficiency advantages in practical computing tasks.

As shown in Figure 5, in the RepVGG module, there are two different convolutional kernels, $3 \times 3$ and $1 \times 1$, during the training phase. This is due to the consistent movement process of convolutional kernels of different sizes when calculated on feature maps. When the model is used for inference, the $1 \times 1$ and $3 \times 3$ convolution kernels can be combined into a single $3 \times 3$ kernel through structural re-parameterization. The specific approach involves padding the surrounding part of the $1 \times 1$ kernel into a $3 \times 3$ form. Based on the additivity principle of convolution kernels of the same size, the padded kernel is added to the original $3 \times 3$ convolution kernel to form a $3 \times 3$ convolution kernel for inference.

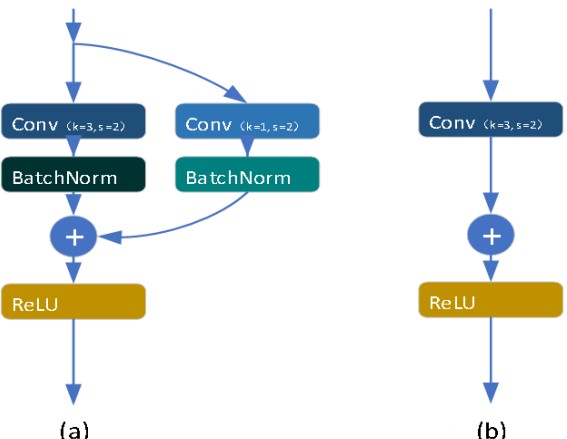

**Figure 5.** RepVGG layer used as stage downsampling: (**a**) RepVGG layer in training; (**b**) RepVGG layer in inference.

*3.2. Neck*

In the proposed network, as shown in Figure 6, the neck's structure is a PAFPN structure network. The neck consists of top-down and bottom-up branches. In the top-down branch, we have top-down layer 1, layer 2, and layer3. The different layers in the top-down branch receive feature maps P1, P2, P3, P4, and P5 from the backbone's stem layer, and stage layer 1, stage layer 2, stage layer 3, and stage layer 4 through the SPPF modules, respectively. The bottom-up branch is composed of three parts: layer 0, layer 1, and layer 2. The input of the bottom-up comes from the output of the top-down branch, as well as the feature map of the backbone's stage layer 4 through SPPF. Their output consists of four feature maps of different sizes, C2, C3, C4, and C5, corresponding to the detection heads of targets of different sizes.

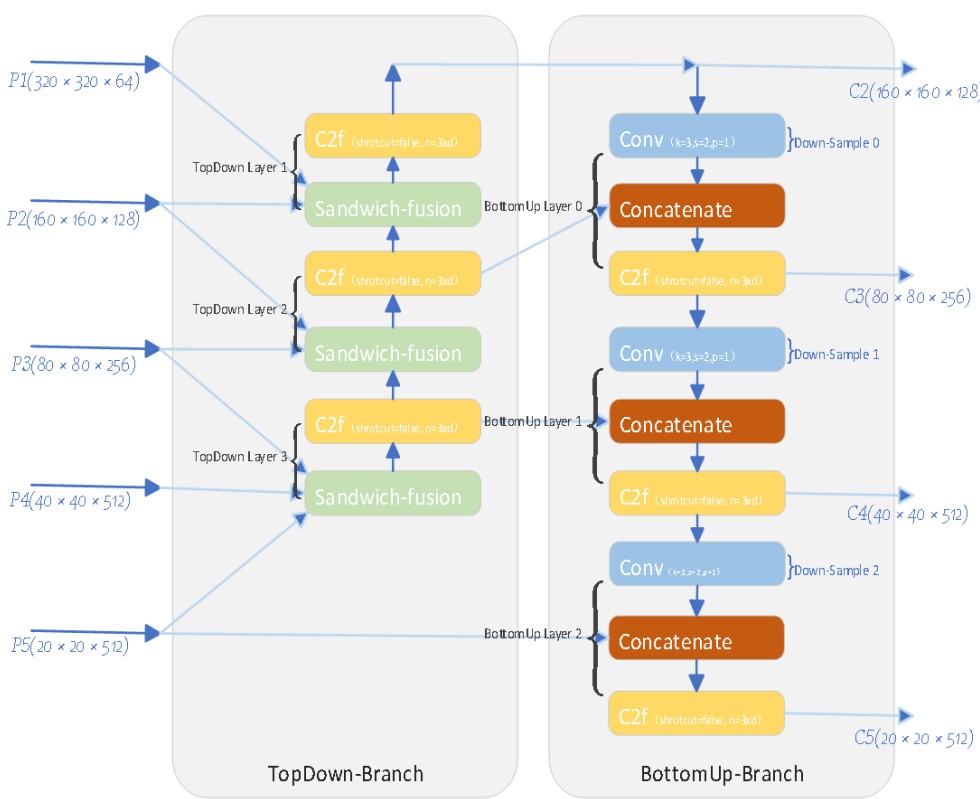

**Figure 6.** The neck of Drone-YOLO (large).

Given that the image data size we need to analyze is relatively large and the objects within these images are comparatively small and dense, following the YOLOv8 strategy of using the three-level PAFPN, suitable for detecting small objects in images, would limit the effectiveness of detecting a large number of tiny and densely overlapping objects within large-sized images. Therefore, we expanded the PAFPN module to a four-layer structure, taking feature maps from the backbone's stem layer and stage layers and outputting four feature maps of the same size to four different detecting heads to detect tiny, small, medium, and large sizes targets.

As shown in Figure 7, we propose sandwich-fusion (SF), a novel fusion module of a three-size feature map, which optimizes the target's spatial and semantic information for detection heads. The module is applied to the neck's top-down layers. The inspiration for this module comes from the BiC model proposed in YOLOv6 3.0 [27]. The input of sandwich-fusion (SF) is shown in the figure, including feature maps from the backbone's lower stage, corresponding stage, and higher stage. The goal is to balance spatial information of low-level features and semantic information of high-level features to optimize the network head's recognition of the target position and classification.

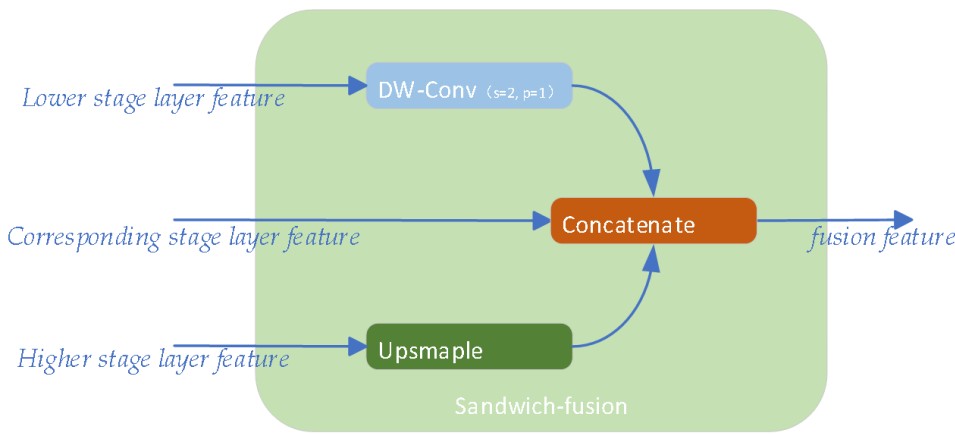

**Figure 7.** Structure of sandwich-fusion (SF).

The sandwich-fusion (SF) module differs from the BiC model of YOLOv6 3.0 in that it implements a depthwise separable convolution layer [40] to extract the feature map from the lower stage layer, which contains the target's most detailed and precise position information. The module concatenates the lower, corresponding, and upsampled high-level feature maps as its output. The subsequent C2f module in the top-down layer extracts this combined information, which is rich in detail regarding the target's classification and spatial positioning.

The depthwise separable convolution is a two-stage convolution, which includes depthwise convolution only for feature maps in a single channel and pointwise convolution only for feature map channels. Compared to convolutional layers with the same-sized kernels, the parameters of depthwise separable convolution are relatively small. This method achieved good results on RepLKNet [41] by expanding the receptive field without increasing numerous parameters.

The PAFPN module is deployed in the improved YOLOv8 as its neck. Its classification information was obtained well through the SPPF layer, but spatial information may be inaccurate. Note that feature maps in the early stages of the backbone are ideal feature sources for spatial information, as these features do not undergo multiple downsamplings and retain relatively detailed positional information. However, the disadvantage of these features is that in the early stage of the backbone, the receptive field of the network is small, and there are some defects in correctly identifying the background and location information from complex texture features. Therefore, in the sandwich-fusion (SF) module, a large kernel depthwise separable convolution is deployed to extract more accurate position information from the early-stage features and filter out irrelevant information related to the detected object.

### 3.3. Proposed Models

In the network neck, the sandwich-fusion (SF) module in the top-down branch of the extended PAFPN structure varies in the sizes of the convolutional kernels used for the underlying depthwise separable convolutional layer, depending on the location of different layers. The specific data are displayed in Table 1:

**Table 1.** Kernel and channel size of depthwise separable convolutional layers in top-down branch of proposed Drone-YOLO (large) Model's neck.

| Position of Module | Module Name | Kernel Size of DWConv | Channels of DWConv |
|---|---|---|---|
| Top-Down layer 1 | Sandwich-fusion | 33 | 32 |
| Top-Down layer 2 | Sandwich-fusion | 15 | 64 |
| Top-Down layer 3 | Sandwich-fusion | 7 | 128 |

Building on the Drone-YOLO (large) model, we employed techniques to create a range of network models with different scales. These techniques include reducing the C2f modules in each stage of the backbone, decreasing the channels of convolutional layers. Consequently, we developed other four distinct scale network models: Drone-YOLO (medium), Drone-YOLO (small), Drone-YOLO (tiny), and Drone-YOLO (nano).

Among these models, Drone-YOLO (large) achieves the highest detection accuracy but exhibits the largest size parameters and highest computational complexity. On the other hand, Drone-YOLO (small) and Drone-YOLO (tiny) possess fewer parameters, deliver satisfactory performance, and operate at faster processing speeds. Notably, when tested on UAV remote sensing images, the tiny model performs exceptionally well in detecting small and tiny targets. It yields results that are comparable to YOLOv8l while utilizing only one-tenth of the latter's parameters. This characteristic makes it highly suitable for deployment on edge computing devices.

For the proposed set of network models, the model parameters for the backbone are presented in Table 2, while the model parameters for the neck are displayed in Table 3.

**Table 2.** Proposed model parameters in the backbone.

| Model Type | Stem | Stage Layer 1 | | Stage Layer 2 | | Stage Layer 3 | | Stage Layer 4 | | SPPF |
|---|---|---|---|---|---|---|---|---|---|---|
| | | RepVGG | C2f | RepVGG | C2f | RepVGG | C2f | RepVGG | C2f | |
| Drone-YOLO (large) | 64 | 128 | $128/n = 3$ | 256 | $256/n = 6$ | 512 | $512/n = 6$ | 1024 | $1024/n = 3$ | 1024 |
| Drone-YOLO (medium) | 48 | 96 | $64/n = 2$ | 192 | $128/n = 4$ | 384 | $256/n = 4$ | 768 | $768/n = 2$ | 768 |
| Drone-YOLO (small) | 32 | 64 | $64/n = 1$ | 128 | $128/n = 2$ | 256 | $256/n = 6$ | 512 | $512/n = 1$ | 512 |
| Drone-YOLO (tiny) | 24 | 48 | $48/n = 1$ | 88 | $88/n = 2$ | 176 | $176/n = 2$ | 344 | $344/n = 1$ | 334 |
| Drone-YOLO (nano) | 16 | 32 | $32/n = 1$ | 64 | $88/n = 2$ | 128 | $128/n = 2$ | 256 | $256/n = 1$ | 256 |

**Table 3.** Proposed model parameters in the neck.

| Model Type | Top-Down Branch | | | | | | Bottom-Up Branch | | |
|---|---|---|---|---|---|---|---|---|---|
| | Top-Down Layer 1 | | Top-Down Layer 2 | | Top-Down Layer 3 | | Bottom-Up Layer 0 | Bottom-Up Layer 1 | Bottom-Up Layer 2 |
| | SF | C2f | SF | C2f | SF | C2f | C2f | C2f | C2f |
| Drone-YOLO (large) | 32 | $128/n = 3$ | 64 | $256/n = 3$ | 128 | $512/n = 3$ | $256/n = 3$ | $512/n = 3$ | $1024/n = 3$ |
| Drone-YOLO (medium) | 24 | $96/n = 2$ | 48 | $192/n = 2$ | 96 | $384/n = 2$ | $192/n = 2$ | $384/n = 2$ | $768/n = 2$ |
| Drone-YOLO (small) | 16 | $64/n = 1$ | 32 | $128/n = 1$ | 64 | $256/n = 1$ | $128/n = 1$ | $256/n = 1$ | $512/n = 1$ |
| Drone-YOLO (tiny) | 16 | $48/n = 1$ | 24 | $88/n = 1$ | 48 | $176/n = 1$ | $88/n = 1$ | $176/n = 1$ | $344/n = 1$ |
| Drone-YOLO (nano) | 8 | $32/n = 1$ | 16 | $64/n = 1$ | 32 | $128/n = 1$ | $64/n = 1$ | $128/n = 1$ | $256/n = 1$ |

Table 2 presents the parameters of modules of the backbone, including the stem, stage layers 1 to 4, and the SPPF. For instance, in the case of Drone-YOLO (large), the stem consists of a convolutional layer with a channel of 64 and a stride of 2. In stage layer 1, the downsampling layer is composed of a RepVGG module with a channel of 128 and a stride of 2. The network component of stage layer 1 comprises 3 cascaded C2f components with a channel of 128.

Table 3 presents the parameters of the modules within the neck, including top-down and bottom-up branches. For instance, in the case of the Drone-YOLO (tiny) network, layer 1 of the top-down branch consists of the sandwich-fusion (SF) module's depthwise separable convolution with a channel of 16, and a single-level C2f module with a channel of 48.

The Drone-YOLO models utilize the same detection decoupled head as YOLOv8, where the classification and regression tasks are separate branches. Each branch of the detection head consists of three cascaded convolutional layers. The first and second layers employ a $3 \times 3$ kernel, while the third utilizes a $1 \times 1$ convolutional kernel. The two branches are responsible for BBox loss and class loss, respectively. The classification task

employs binary cross entropy (BCE) loss, while the regression task utilizes a combination of complete IoU (CIoU) loss and distribution focal loss (DFL).

## 4. Experiments and Results

### 4.1. Dataset and Experiments Environment

The VisDrone2019 dataset [42] was collected, annotated, and organized byby the AISKYEYE team from Tianjin University's machine learning data mining laboratory. The object detection dataset of the VisDrone2019 consists of images and corresponding annotation files, including 6471 images in the training set, 548 images in the validation set, 1610 images in the test set, and 1580 images in the competition set. In the dataset, the images sizes varied from $2000 \times 1500$ to $480 \times 360$. Because it is taken from the perspective of the drone, the image is quite different from the image taken by the ground personnel, such as MS-COCO and VOC2012 in the photo angle, image content, background, and ambient illumination. There are sample images of VisDrone2019 shown in Figure 8. The scene consisting of dataset images is extensive, covering streets, squares, parks, schools, residential communities, etc. The illumination conditions for images include daytime with sufficient lighting conditions, nighttime with insufficient lighting conditions, cloudy, high-intensity light, and glare conditions. The object types annotated in the image include 10 types: pedestrians, people, bicycles, cars, vans, trucks, tricycles, awning-tricycle, buses, and motors.

In the experiment, we used Ubuntu 20.04 as the operating system with Python 3.8, PyTorch 1.16.0, and Cuda 11.6 as the desktop computational software environment. The experiment utilized NVIDIA 3080ti graphics cards as hardware. The implementation code of the neural network was modified based on the Ultralytics 8.0.105 version. The hyperparameters used during the training, testing, and validation of the experiment remained consistent. The training epoch was set at 300, and the images inputted into the network were rescaled to $640 \times 640$. In the results listed below, all the YOLOv8 and our proposed Drone-YOLO networks have detection results from our experiments. In these experiments, none of these networks used pre-training parameters. The remaining data come from relevant cited papers.

In the embedded application experiment, we used the NVIDIA Tegra TX2 as the experimental environment, which has a 256-core NVIDIA Pascal architecture GPU, providing a peak computing performance of 1.33 TFLOPS, and 8GB of memory. The software environment was Ubuntu 18.04 LTS operating system, NVIDIA JetPack 4.4.1, CUDA 10.2, and cuDNN 8.0.0.

### 4.2. Experiment Metrics

The experiments evaluate proposed methods in terms of detection performance and model parameter size. The experiment metrics include precision ($P$), recall ($R$), average precision ($AP$), mean average precision ($mAP$), and million parameters (M) for the network parameter size.

Precision ($P$) is the proportion of correctly predicted targets among all detected targets. It is calculated through Equation (1), where $TP$ represents the correct prediction targets, and $FP$ represents the incorrect prediction targets.

$$P = \frac{TP}{(TP + FP)} \tag{1}$$

Recall ($R$) is the proportion of correct detected targets among all existing targets. It is calculated through Equation (2), where $FN$ represents targets that exist but have not been correctly detected.

$$R = \frac{TP}{(TP + FN)} \tag{2}$$

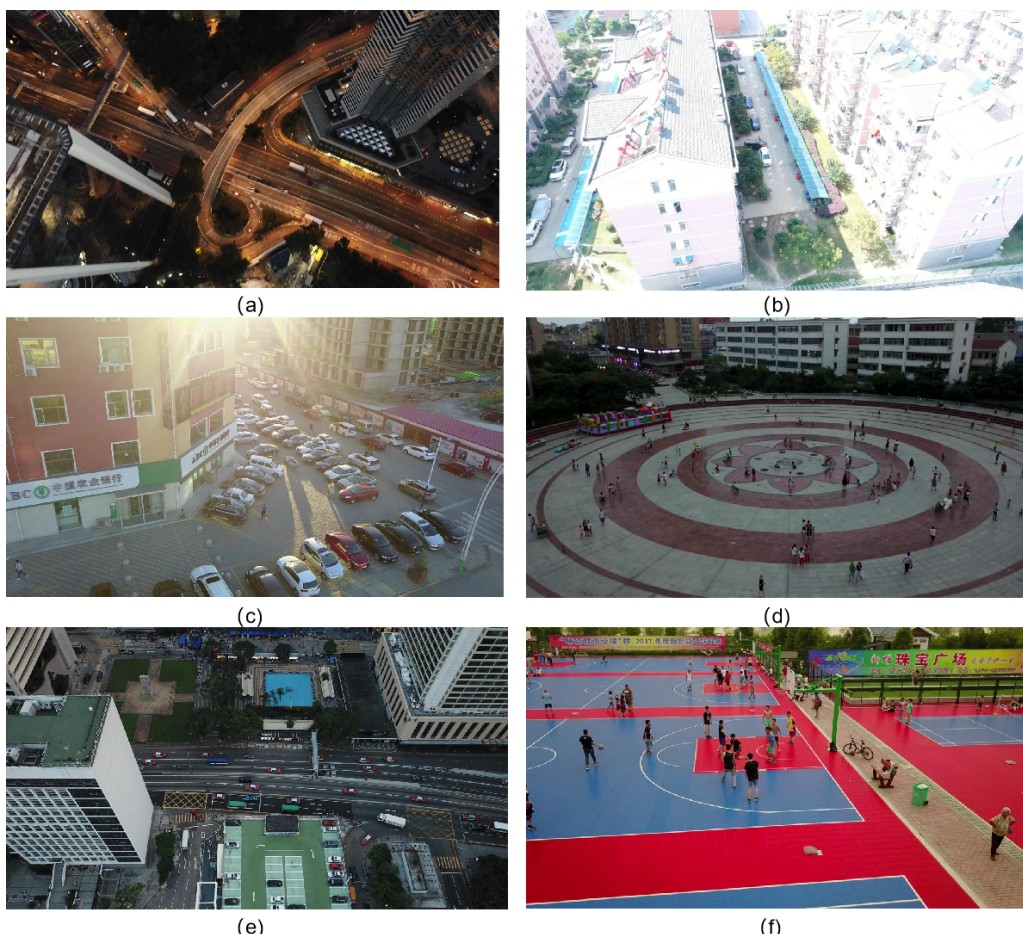

**Figure 8.** Samples of VisDrone2019: (**a**) high-altitude aerial view of urban roads at night with low illumination conditions; (**b**) residential area images under high- intensity daytime lighting conditions; (**c**) urban road intersections under glare conditions; (**d**) city square scene from a wide view under cloudy conditions; (**e**) aerial view image of an urban road area under cloudy conditions; (**f**) daytime low-altitude sports stadium scene.

The average precision ($AP$) represents the area enclosed by the curve formed by the precision and recall. It is calculated through Equation (3). Our metrics include three different average precision indicators: $AP_{0.5}$, $AP_{0.75}$, and $AP_{0.95}$. For $AP_{0.5}$, in order to evaluate bounding box prediction as true, the intersection of the union score (IoU) between the predicted and the annotated bounding box must be higher than 0.50. For $AP_{0.75}$, bounding box predictions with IoU scores higher than 0.75 are considered to be true. For $AP_{0.95}$, we calculate the average precision values of different IoU scores that vary within the range of 0.50:0.05:0.95, and then use the average of these calculated average precision values.

$$AP = \int_0^1 p(r)dr \tag{3}$$

The mean average precision ($mAP$) is the average accuracy of all types of samples. It is calculated through Equation (4). Our metrics include three different mean average precision indicators: $mAP_{0.5}$, $mAP_{0.75}$, and $mAP_{0.95}$.

$$mAP = \frac{1}{k}\sum_{i=1}^{k} AP_i \tag{4}$$

### 4.3. Comparison with the Baseline Methods

Our experimental results were compared with the results of other methods published on this dataset throughout the years, including faster R-CNN [6], RetinaNet [43], cascade R-CNN [44], CenterNet [45], improved-YOLOv4 [46], MSA-YOLO [29], TPH-YOLOv5 [47], and and Li et al. [34]. These methods, as well as YOLOv8-s and YOLOv8-l, are the baseline methods in this experiment.

The experimental results of the comparison methods shown in Figure 9 were tested on the VisDrone2019-test dataset by scaling the input image size to $640 \times 640$. The metric for the evaluated result is $AP_{0.5}$ for each object type, and $mAP_{0.5}$ for all type objects. The proposed Drone-YOLO (large) achieved the best result in $mAP_{0.5}$, and was the best in seven different type objects. Compared with TPH-YOLOv5 [47], which performed best in baseline methods, our Drone-YOLO (large) increased the value of $mAP_{0.5}$ by 9.1%. Drone-YOLO (medium) and Drone-YOLO (small) with smaller parameter sizes also performed well compared to baseline methods.

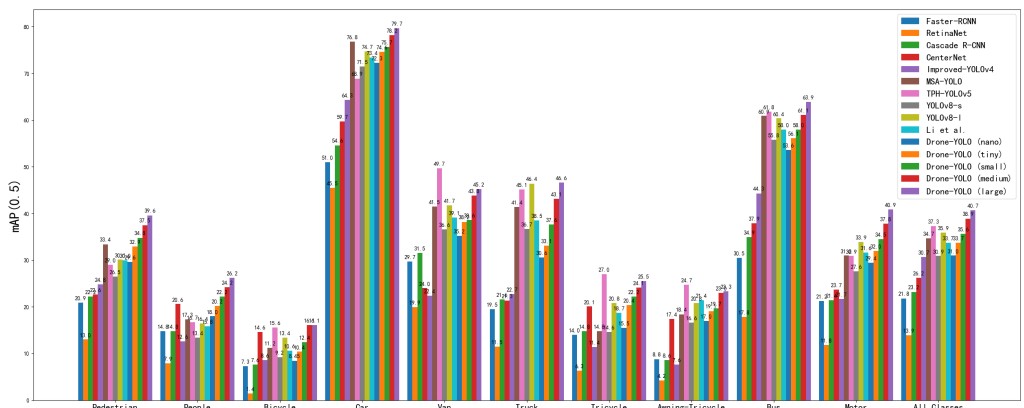

**Figure 9.** Experimental results for all type object detection on the VisDrone2019-test.

Table 4 shows the performance of the proposed methods and the baseline methods on the VisDrone2019-test dataset and model parameter size. In our proposed models, both Drone-YOLO (nano) and Drone-YOLO (tiny) are models with few parameters. However, Drone-YOLO (tiny) only used 55.4% (regarding the size) of the baseline method of Li et al.'s [34] parameters and achieved the same result on $mAP_{0.5}$.

**Table 4.** Experimental results on the VisDrone2019-test.

| Method | $mAP_{0.5}$ | $mAP_{0.75}$ | $mAP_{0.95}$ | Parameters |
|---|---|---|---|---|
| MobileNetv2-SSD | 4.2 | - | - | 3.94 M |
| YOLOv4-s | 26.5 | - | - | 9.12 M |
| YOLOv5-s | 31.0 | - | - | 9.12 M |
| YOLOv5-m | 33.7 | - | - | 25.1 M |
| YOLOX-s | 26.3 | - | - | 8.94 M |
| YOLOv7-tiny | 26.8 | - | - | 6.02 M |
| YOLOv8-s [20] | 30.9 | - | - | 11.1 M |
| YOLOv8-l [20] | 35.9 | 21.8 | 21.2 | 76.7 M |
| Li et al. [34] | 33.7 | - | - | 9.66 M |
| Drone-YOLO (nano) | 31.0 | 17.6 | 17.5 | 3.05 M |
| Drone-YOLO (tiny) | 33.7 | 19.4 | 19.1 | 5.35 M |
| Drone-YOLO (small) | 35.6 | 20.6 | 20.4 | 10.9 M |
| Drone-YOLO (medium) | 38.9 | 22.9 | 22.5 | 33.9 M |
| Drone-YOLO (large) | 40.7 | 24.3 | 23.8 | 76.2 M |

Table 5 shows the performance of the proposed methods and the baseline methods on the VisDrone2019-val dataset and model parameter size. Our proposed Drone-YOLO (large) performs best on $mAP_{0.95}$, while MS-YOLOv7 [31] performs best on $mAP_{0.5}$.

**Table 5.** Experimental results on VisDrone2019-val.

| Method | $mAP_{0.5}$ | $mAP_{0.75}$ | $mAP_{0.95}$ | Parameters |
|---|---|---|---|---|
| YOLOv7-l [19] | 47.1 | - | 26.4 | 71.4 M |
| YOLOv8-l [20] | 43.7 | 27.7 | 26.9 | 76.7 M |
| ACAM-YOLO [30] | 49.5 | - | 29.6 | 15.9 M |
| MS-YOLOv7 [31] | 53.1 | - | 31.3 | 79.7 M |
| Li et al. [34] | 42.2 | - | - | 9.66 M |
| EdgeYOLO [37] | 44.8 | 26.2 | 26.4 | 40.5 M |
| Drone-YOLO (nano) | 38.1 | 22.8 | 22.7 | 3.05 M |
| Drone-YOLO (tiny) | 42.8 | 26.2 | 25.6 | 5.35 M |
| Drone-YOLO (small) | 44.3 | 27.7 | 27.0 | 10.9 M |
| Drone-YOLO (medium) | 48.6 | 31.1 | 30.1 | 33.9 M |
| Drone-YOLO (large) | 51.3 | 33.2 | 31.9 | 76.2 M |

In the embedded environment, we evaluated the performance of our proposed methods, Drone-YOLO (nano), and Drone-YOLO (tiny), as well as the comparative methods, YOLOv5-n, YOLOv5-s, YOLOv8-n, and YOLOv8-s, on the VisDrone2019-val dataset for the object detection task. The evaluations were conducted on the NVIDIA Tegra TX2 platform. As shown in Table 6, Drone-YOLO (tiny) achieved the highest detection accuracy on the dataset.

**Table 6.** Experimental results on the NVIDIA Tegra TX2, VisDrone2019-val dataset.

| Method | $mAP_{0.5}$ | $mAP_{0.95}$ | Average Inference Time per Image | Parameters |
|---|---|---|---|---|
| YOLOv5-n [16] | 33.1 | 19.2 | 27.4 ms | 2.50 M |
| YOLOv5-s [16] | 39.3 | 23.4 | 56.8 ms | 9.12 M |
| YOLOv8-n [20] | 34.0 | 19.8 | 30.8 ms | 3.00 M |
| YOLOv8-s [20] | 39.5 | 23.5 | 62.1 ms | 11.1 M |
| Drone-YOLO (nano) | 38.1 | 22.7 | 127.4 ms | 3.05 M |
| Drone-YOLO (tiny) | 42.8 | 25.6 | 189.5 ms | 5.35 M |

### 4.4. Ablation Experiment

In ablation experiments, the baseline method for the comparison is the YOLOv8-l model. We added a small-sized object detection head (YOLOv8-l + header), a sandwich fusion module (YOLOv8-l + sandwich + header), and RepVGG modules in the backbone as the downsampling layers (YOLOv8-l + sandwich + header + RepVGG). After the above three stages of improvement, we developed our proposed Drone-YOLO (large) model. Each model measured multiple metrics in the VisDrone2019-Val dataset. In this experiment, all hyperparameters remained unchanged. During our training process, the input image size was $640 \times 640$, the batch size of all models was set to 8, and the training period was 300 epochs.

The experimental results are shown in Table 7. From the results, the most significant improvement of the YOLOv8-l model was the addition of small-sized object detection heads, which improved the most on the strictest $mAP_{0.95}$ index, reaching 19.3%. This indicates the importance of detecting small-sized targets from spatial-rich feature maps in this task. After adding the sandwich fusion module, the improvement measured on the dataset was the least significant, with a slight decrease in the $mAP_{0.75}$ indicator. However, the $mAP_{0.5}$ and the strictest $mAP_{0.95}$ indicators slightly increased. When using the RepVGG

structure as the downsampling module for the backbone part, there was some improvement in this result, with an increase of 1.2% based on the $mAP_{0.5}$ indicator.

**Table 7.** Ablation experiment result in VisDrone2019-val.

| Method | $mAP_{0.5}$ | $mAP_{0.75}$ | $mAP_{0.95}$ |
|---|---|---|---|
| YOLOv8-l | 43.7 | 27.7 | 26.4 |
| YOLOv8-l + header | 50.5 (↑ 15.6%) | 32.7 (↑ 18.1%) | 31.5 (↑ 19.3%) |
| YOLOv8-l + sandwich + header | 50.7 (↑ 0.4%) | 32.5 (↓ 0.6%) | 31.6 (↑ 0.3%) |
| YOLOv8-l + sandwich + header + RepVGG | 51.3 (↑ 1.2%) | 33.2 (↑ 2.1%) | 31.9 (↑ 0.9%) |

*4.5. Visualization*

The following is an analysis of the differences in the object detection results between the proposed Drone-YOLO (large), Drone-YOLO (tiny), and YOLOv8-l in images from different scenes, lighting conditions, shooting positions, and object types.

As shown in Figure 10, in the middle right part of the image, there are five cars driving on the street, marked with red boxes. The first car just arrived at the foot of the pedestrian overpass, and two bright lights can be seen shining on the ground. For these five cars, Drone-YOLO (large) detected two of them, while the other two methods failed to detect any of them. In the middle-left part of the image, there are two cars driving on the road, marked with red boxes. Drone-YOLO (large) detected two vehicles with a high degree of stability. Drone-YOLO (tiny) also detected two vehicles but with slightly lower durability. YOLOv8-l did not detect either vehicle.

As shown in Figure 11, the upper part of the picture, circled by a large red box, is a main road with many cars driving on the road. Due to the high altitude and distance of aerial photography, the size of the car in the image is relatively tiny. YOLOv8-l only detected three vehicles in this area of the image. Drone-YOLO (large) and Drone-YOLO (tiny) both detected most targets. In the center of the picture, there are two cars driving on the auxiliary road. YOLOv8-l did not detect these two cars either; however, both Drone-YOLO (large) and Drone-YOLO (tiny) detected these two cars. On the left side of the image, there is a car on a small road behind a building. For the target exposed in the building gap, neither YOLOv8-l nor Drone-YOLO (tiny) detected it, but Drone-YOLO (large) detected the car.

As shown in Figure 12, in a low-illumination night environment, there are two motorcycles and two cars driving under the overpass in the red box on the left side of the image. Drone-YOLO (large) detected one car, two motorcycles, and pedestrians on the motorcycles. YOLOv8-l detected one car and one motorcycle. Drone-YOLO (tiny) only detected one car.

As shown in Figure 13, the central part of the image is a street scene with sufficient daytime lighting conditions. The drone is about 60 m above the ground, and the street area circled in the big red box is far from the drone, with a large number of cars driving on the road. YOLOv8-l can only detect a small number of cars. Both Drone-YOLO (large) and Drone-YOLO (tiny) can detect many of these cars.

As shown in Figure 14, the image scene is a traffic intersection. The drone is located relatively low from the ground, and there is a significant glare on the captured road surface. There is an entrance to a shopping mall on the corner of the street, where crowds enter and exit. There is a larger red box here. YOLOv8-l detected one person in this area, Drone-YOLO (large) detected many people, and Drone-YOLO (tiny) detected fewer people, but more than YOLOv8-l. The traffic post highlighted in the red box in the figure was mistakenly detected by YOLOv8-l as a tricycle, while Drone-YOLO (large) did not show any false alarms. Drone-YOLO (tiny) mistakenly detected tricycles and people.

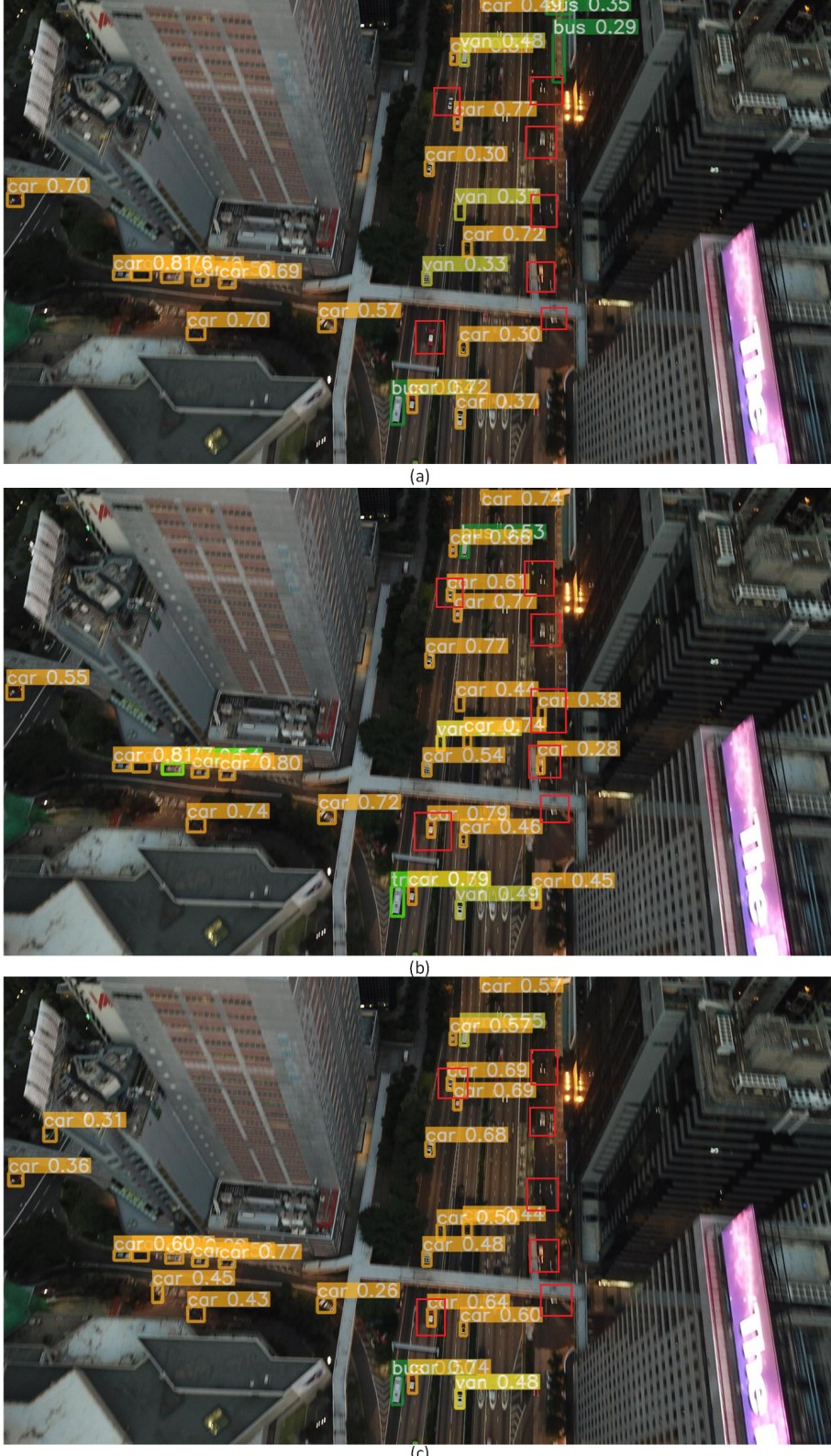

**Figure 10.** A daytime urban street scene with insufficient lighting conditions. The image was captured from an elevated position exceeding 100 meters above the ground. The primary detection objects include cars, buses, and vans. (**a**) Results of YOLOv8-l; (**b**) results of Drone-YOLO (large); (**c**) results of Drone-YOLO (tiny).

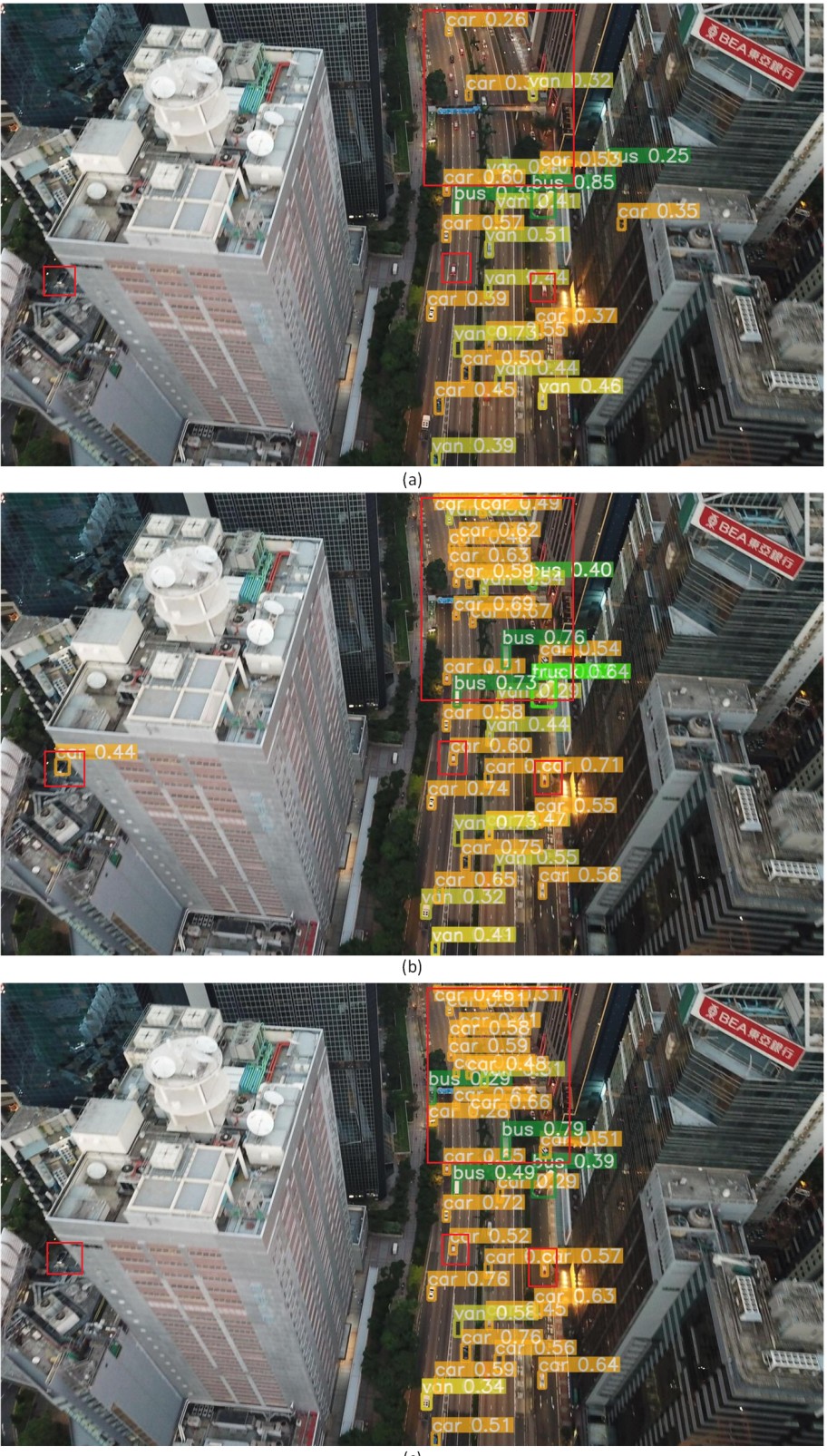

**Figure 11.** A daytime urban street scene with normal lighting conditions. The image was captured from an elevated position exceeding 100 meters above the ground. The primary detection objects include cars, buses, and vans. (**a**) Results of YOLOv8-l; (**b**) results of Drone-YOLO (large); (**c**) results of Drone-YOLO (tiny).

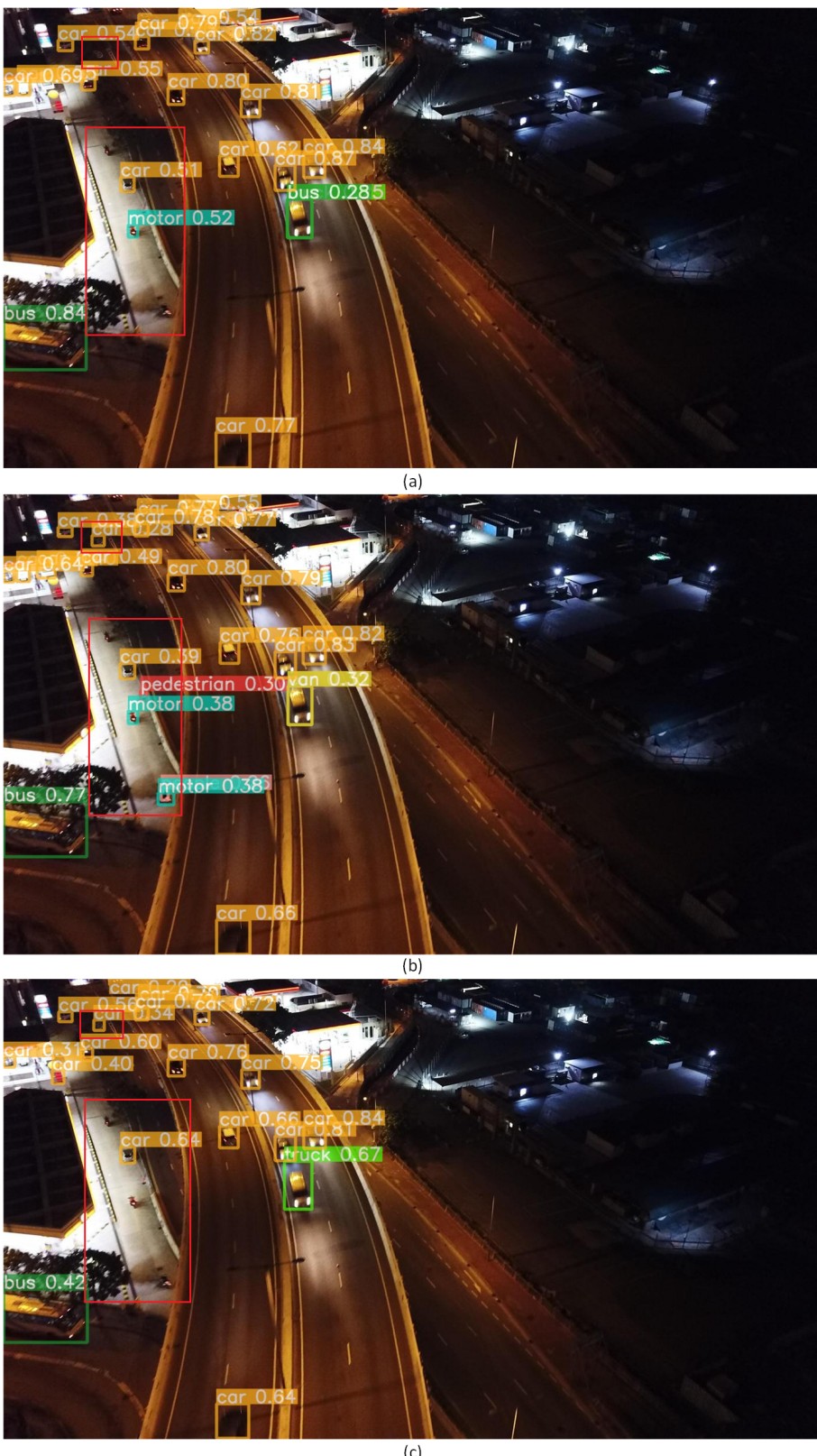

**Figure 12.** A nighttime suburban street scene with insufficient lighting conditions. The image was captured from an elevated position that is approximately 30 meters above the ground at an oblique angle. The primary detection objects include cars, vans, motorcycles, and pedestrians. (**a**) Results of YOLOv8-l; (**b**) results of Drone-YOLO (large); (**c**) results of Drone-YOLO (tiny).

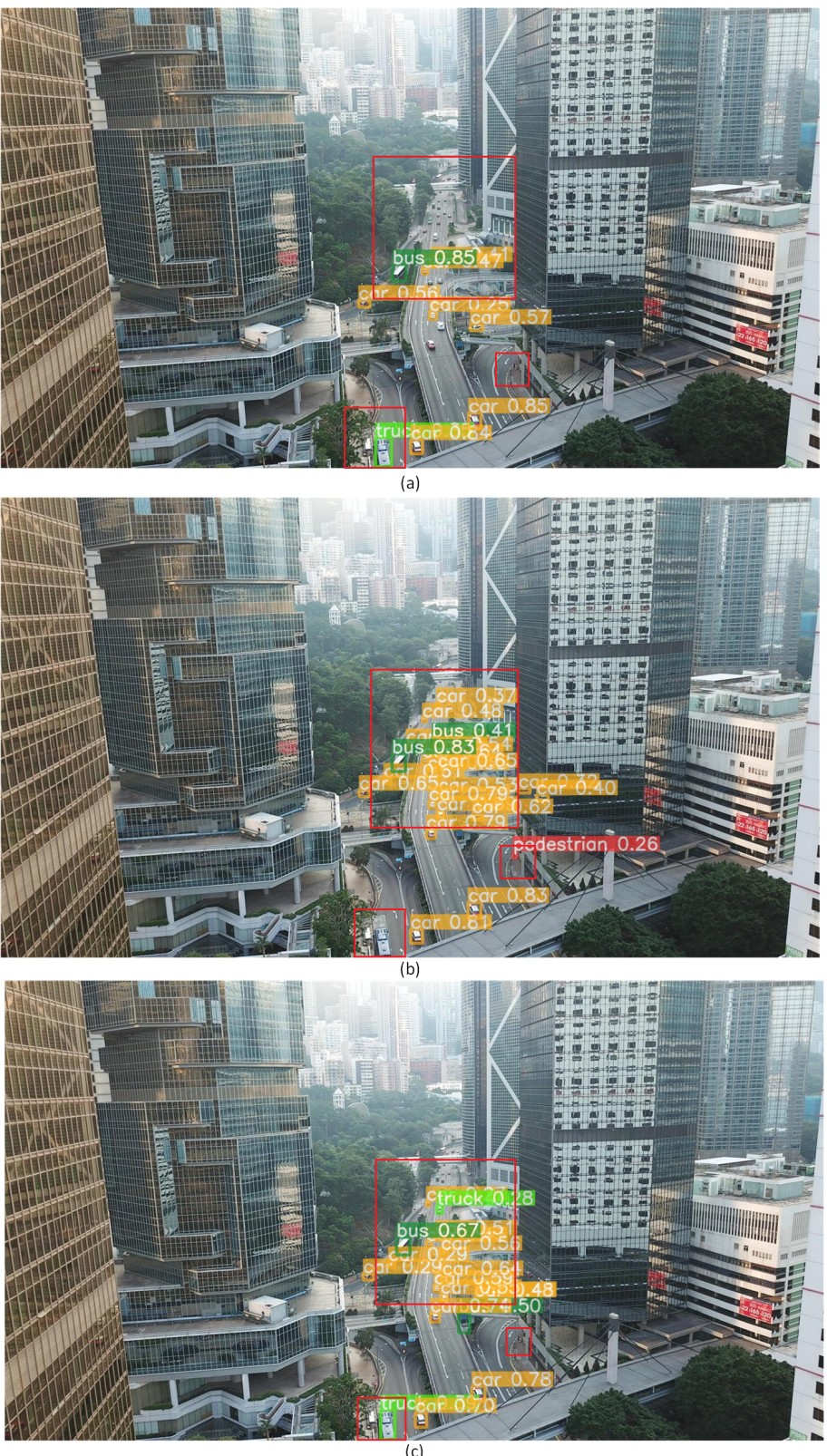

**Figure 13.** A daytime urban street scene with ample lighting conditions. The image was captured from an elevated position approximately 60 meters above the ground at an oblique angle. The primary detection objects include cars, buses, and pedestrians. (**a**) Results of YOLOv8-l; (**b**) results of Drone-YOLO (large); (**c**) results of Drone-YOLO (tiny).

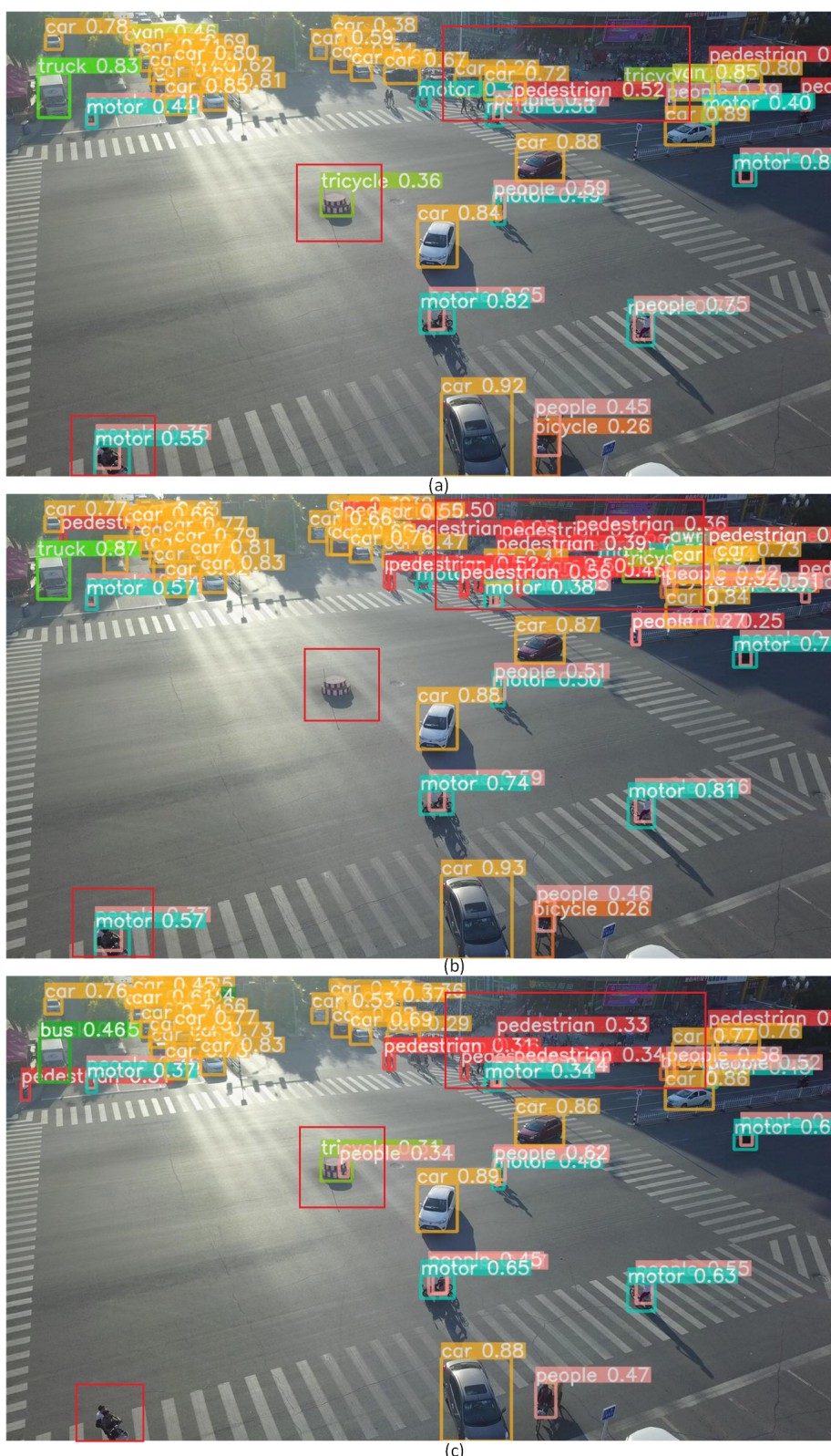

**Figure 14.** A morning street scene with ample lighting conditions. The image was captured from a slightly inclined position approximately 15 meters above the ground. The primary detection objects include cars, pedestrians, motorcycles, people, vans, and trucks. (**a**) Results of YOLOv8-l; (**b**) results of Drone-YOLO (large); (**c**) results of Drone-YOLO (tiny).

### 5. Conclusions

In this article, we propose multi-scale drone image object detection algorithms, called Drone-YOLO, based on the YOLOv8 model. These algorithms are designed to address the specific challenges associated with drone image object detection. Given the large number of drone image scenes and the relatively small number of detection objects, we made improvements to the neck of the YOLOv8 model. We introduced a three-layer PAFPN structure to enhance the detection of small-sized objects based on $160 \times 160$ feature maps. This improvement significantly enhances the algorithm's ability to detect small-sized targets. Furthermore, we incorporated the sandwich-fusion module in each layer of the neck's up–down branch. This structure allows the fusion of network features with low-level features that contain rich object spatial information. We achieve this fusion through the use of depthwise separable evolution, which incurs small parameters and provides a large receptive field. In the backbone of the network, we utilize the RepVGG module as the downsampling layer. The RepVGG module enhances the network's capacity to learn multi-scale features and yields improved detection results compared to the convolutional layer.

In our experiments, in the ViSDrone2019-test dataset, our proposed Drone-YOLO (large) outperforms baseline methods in class object detection in $mAP_{0.5}$, and performs the best in 7 out of 10 classes in $AP_{0.5}$ metrics. In the VisDrone2019-val dataset, our approach outperforms other methods in terms of the $mAP_{0.95}$ metrics. Drone-YOLO (tiny), with only 5.35 M parameters, achieves close results to the baseline method with 9.66 M parameters in $mAP_{0.5}$ metrics on the VisDrone2019-test dataset. On the VisDrone2019-val dataset, it surpasses the baseline method in $mAP_{0.5}$ metrics. It is worth noting that our network improvement methods do not incorporate currently popular attention mechanisms, indicating potential for further enhancements based on task-specific variations.

The proposed methods consist of models with different parameter sizes, where models with larger parameters are suitable for post-flight object detection on desktop computing platforms. Models with smaller parameters have undergone testing on the NVIDIA Tegra TX2 platform and have demonstrated highly accurate results. Considering the computational capabilities of NVIDIA's recently released platforms, such as Jetson Xavier NX and Jetson AGX Xavier, our proposed approach is well-suited for performing high-accuracy object detection tasks on high-performance embedded platforms integrated with UAVs.

**Funding:** This research received no external funding.

**Data Availability Statement:** Not applicable.

**Conflicts of Interest:** The authors declare no conflict of interest.

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
