# Peer review of "Drone-YOLO: An Efficient Neural Network Method for Target Detection in Drone Images"

_drones, doi:10.3390/drones7080526_

Round 1

Reviewer 1 Report

Thanks for providing me an opportunity to review the manuscript titled Drone-YOLO: An Efficient Neural Network Method for Target Detection in Drone Images, submitted to Drones.

- Drone-YOLO: An Efficient Neural Network Method for Target Detection in Drone Images, drones-2499712-peer-review-v1

The author provided quite interesting methodology and results about pattern recognitions collected from UAV surveying and data acquisitions with deep learning algorithms.

However, the manuscript should be improved as followings:

1. Flowcharts and diagrams should be enhanced. Most of the diagrams and pictures were hard to understand - too much information as displayed.

2. A part of the introduction in the manuscript should be rewritten. There are many misspelling and repetitive words. In particular, three main contributions at the end of the section are rewritten. 

3. In table 4, what does Tri, Awn-Tri or Motor mean? I suggest Table 4 converting into a vertical bar graph or reducing a number of factors. I understand that an author wanted to highlight better performance of 5 different types of Drone-YOLO approaches over other results from object recognition methods. However, there is too much information in Table 4. 

Moreover, compared with previous peer-reviewed papers as below, an author should have to significantly improve how to demonstrate results, to organize structures and to write effectively most paragraphs etc.

Li, Y. et al. A Modified YOLOv8 Detection Network for UAV Aerial Image Recognition. Drones 2023, 7, 304. https://doi.org/10.3390/drones7050304

Luo, X. et al. YOLOD: A Target Detection Method for UAV Aerial Imagery. Remote Sens. 2022, 14, 3240. https://doi.org/10.3390/rs14143240

.

Author Response

Dear Reviewer:

Thank you for your review, thank you for your valuable feedback, and thank you for your efforts. I have revised this article one by one based on your feedback.

1. Figures 3 and 6 in this article may not have been clear enough, and the characters in the images may be small, which has been modified. The network parameters in the current graph are relatively easy to read.

2. The Introduction section and Contribution section of this article have been rewritten based on your feedback.

3. According to your opinion, the original Table 4 of this article has been changed to the ground bar model, which can visually display the differences in the results of various methods. The original "Tri" and "Awn Tri" in Table 4 refer to tricycle and Awning tricycle

The revised manuscript is attached, and all the parts you suggested to modify have been highlighted.

Yours sincerely

Zhengxin Zhang

Reviewer 2 Report

1-Page 4-5, describe the existing gap in literature and how you are addressing it.

2-Please explain how you envision the result of this study will be used and applied in practice.

Author Response

Dear Reviewer:

Thank you for your review and valuable feedback on the revisions. Thank you for your efforts.

  1. At the end of the second paragraph of this article, we have added the lack of relevant methods for target detection in unmanned aerial vehicle remote sensing images, as well as the improvement methods for our method.

  1. At the end of the conclusion paragraph of this article, it is added that the proposed method can be applied in practical scenarios.

Yours sincerely

Zhengxin Zhang

Reviewer 3 Report

This manuscript improves the target detection method for drone images and proposes multiple size network structures for deployment. The manuscript is of high quality, with sufficient theory and experiments, especially the introduction with clear logic and comprehensive review. Suggest accept after minor revision.

1. The full manuscript should reduce first person descriptions. For objective description, Passive voice should be used.

2.Figure 3 should increase clarity and expand the textual content in the image.

3. It can properly supplement the computing power of edge computing platform for networks of different sizes proposed in the paper and deployed to UAVs. This provides a reference for engineering personnel to use this method (selecting which computers can carry the proposed different networks).

4. If possible, comparative experiments should be added to the experimental section of section 4.5. Verify the effectiveness of the proposed method.

Author Response

Dear Reviewer:

Thank you for your review and valuable feedback on the revisions. Thank you for your efforts.

Based on your review comments, the manuscript has been revised one by one, and the parts where you provided comments have been highlighted.

1. Based on your feedback, I have rewritten the first part of this article. I have annotated the first person descriptions you mentioned as a contribution to the first part of this article. I have rewritten it according to your opinion.

2. The resolution of Figures 3 and 6 in the article has been regenerated, and the characters in the images have been enlarged, hoping to be easier to read.

3. Based on your feedback, we have built the NVIDIA tegra TX2 embedded platform and conducted experiments to obtain data, which is highlighted in section 4.3 of the article.

4. Further experimental results in section 4.5 were not updated. The reason is that in the proposed model, Drone YOLO (large) and Drone YOLO (tiny) are the two models with the largest and smallest parameters in the series, and their running results can basically represent the upper and lower limits that can be achieved by networks with other parameter sizes, except for the Drone YOLO (nano) model with the least parameters in this series. The demonstrated images also include detection results of different types of targets under different lighting conditions. And the size of a single image is also relatively large. If the results of five different network models are posted, each image will be very small, and the differences in details will be difficult to see clearly. From the perspective of typesetting, this method is currently relatively optimal. Of course, if you think more result images are better without considering the image size, I will list all the images.

Your sincerely

Zhengxin Zhang

Round 2

Reviewer 1 Report

.

.